# Cross-Component Transferable Transformer Pipeline Obeying Dynamic Seesaw for Rotating Machinery with Imbalanced Data

**DOI:** 10.3390/s23177431

**Published:** 2023-08-25

**Authors:** Binbin Xu, Boquan Ma, Zheng Yang, Fei Chen, Xiaobing Li

**Affiliations:** 1Sino-German College of Intelligent Manufacturing, Shenzhen Technology University, Shenzhen 518118, China; xubinbin@sztu.edu.cn (B.X.); maboquan1025@163.com (B.M.); 2School of Mechanical and Aerospace Engineering, Jilin University, Changchun 130025, China; yangzhengjlu@163.com; 3Guangdong Provincial Key Laboratory of Electronic Information Products Reliability Technology, Guangzhou 511370, China

**Keywords:** rotating machinery, fault diagnosis, transformer, transfer learning

## Abstract

Due to the lack of fault data in the daily work of rotating machinery components, existing data-driven fault diagnosis procedures cannot accurately diagnose fault classes and are difficult to apply to most components. At the same time, the complex and variable working conditions of components pose a challenge to the feature extraction capability of the models. Therefore, a transferable pipeline is constructed to solve the fault diagnosis of multiple components in the presence of imbalanced data. Firstly, synchrosqueezed wavelet transforms (SWT) are improved to highlight the time-frequency feature of the signal and reduce the time-frequency differences between different signals. Secondly, we proposed a novel hierarchical window transformer model that obeys a dynamic seesaw (HWT-SS), which compensates for imbalanced samples while fully extracting key features of the samples. Finally, a transfer diagnosis between components provides a new approach to solving fault diagnosis with imbalanced data among multiple components. The comparison with the benchmark models in four datasets proves that the proposed model has the advantages of strong feature extraction capability and low influence from imbalanced data. The transfer tests between datasets and the visual interpretation of the model prove that the transfer diagnosis between components can further improve the diagnostic capability of the model for extremely imbalanced data.

## 1. Introduction

Rotating machines (RM) are widely used in intelligent equipment such as computerized numerical control (CNC) machines, aircraft engines, wind turbines, etc. Economic losses and the closure of some facilities are the results when RM fails or stops. More specifically, most RM failures are caused by typical components such as bearings, gears, motors, etc. These components operate in a complex environment and have different fault classes. Therefore, timely and accurate fault diagnosis for these typical components can reduce unnecessary malfunctions and downtime, which is essential for improving the reliability and safety of RM.

In general, the main methods of fault diagnosis consist of model-driven methods and data-driven methods. The model-driven method is needed to analyze the fault mechanism based on experience and prior knowledge [1]. In contrast, with the widespread development of deep learning (DL) in multiple research fields [2,3], extensive research has been conducted on data-driven fault diagnosis combined with signal processing technology, which can achieve end-to-end fault diagnosis without requiring extensive expertise [4,5]. Furthermore, fault diagnosis based on contactless sensing data has begun to be studied. Li et al. [6] provided a new contactless health monitoring and fault diagnosis method by collecting visual data on vibration through event-based cameras for the first time. However, data-driven methods as presented by DL require huge amounts of labeled data and balanced sample data between different classes. However, it is difficult to collect adequate and balanced data because RM is usually in normal operation in modern manufacturing [7]. Imbalanced samples can cause the model to excessively learn features from healthy samples. resulting in “underreporting”, which will reduce the accuracy and reliability of the model and cause losses to the production safety of the enterprise. Therefore, it is of practical importance to explore the methods of fault diagnosis of RM in the presence of imbalanced data.

Numerous methods and strategies have been developed for dealing with the imbalanced data from RM, generally divided into model-based and data-based methods. Model-based methods learn features from the imbalanced samples by constructing an algorithm model. Li et al. [8] constructed a cost-sensitive multi-decision tree algorithm, which increases the fault cost of learning samples from minority classes and makes the model more sensitive to minority class data. Sun et al. [9] proposed an automatic imbalance diagnosis method based on a Bayesian optimizer that optimizes the parameters of oversampling models and classifier models through a hierarchical parameter space, achieving diagnostic tasks under various imbalance ratios. Currently, designing a model structure to enhance its feature extraction ability is a more intelligent method. Wang [10] proposed a normalized softmax loss with adaptive angle margin to supervise neural networks learning imbalanced data. However, it is difficult to formulate the cost strategy and improve the model’s ability to learn features. Data-based methods mainly refer to resampling techniques, including under-sampling methods (USM) for multi-class samples and over-sampling methods (OSM) for the few class samples, all designed to balance the class distribution [11]. Tang et al. [12] used extreme gradient boosting feature selection and improved whale optimization random forest to diagnose the fault of a wind turbine gearbox by under-sampling the normal data. Although the influence of imbalanced data on the model is eliminated to some extent by USM, some feature information from the normal data was lost in the process. In contrast, OSM is more commonly used because it expands the samples of a few classes based on the existing data. Zhang et al. [13] proposed a weighted minority OSM and used an improved deep auto-encoder (AE) as the backbone of feature extraction, which can avoid generating incorrect or unnecessary samples. Wei et al. [14] used k-nearest neighbors to filter out noisy points from OSM-generated samples and made a transition from multiple binary class imbalances to multiple class imbalances for RM. In traditional OSM, such as the synthetic minority over-sampling technique (SMOTE) [15], the pseudo-samples generated by OSM have poor generalization and some noisy points. Although the above-mentioned improved OSM overcomes the traditional problems, there is still the problem that the sampling distribution features cannot be learned automatically. Moreover, the generative adversarial network (GAN) [16] has been widely used for imbalanced data because it can compensate for imbalanced data by generating pseudo-samples. Mao et al. [17] used the spectrum data of the bearing vibration signal to generate samples with few classes using GAN and a stacked denoising model AE to perform fault diagnosis. Zhao et al. [18] improved the accuracy and diversity of the generated data by using an improved GAN, which combined AE and an online sample filter, and then introduced an additional classifier to train 2D images transformed by wavelet transform from the bearing vibration signal. Zareapoor et al. [19] proposed the minority oversampling generative adversarial network (MoGAN), which not only produces high-quality patterns with few classes but also enables the discrimination of pseudo-patterns. The samples generated by GAN and its derivatives have the same distribution as the original samples, but this method is still limited by the quality of the original samples.

Recently, deep transfer learning (DTL) has been used in the fault diagnosis of RM to overcome the overdependence on the original samples [20], which is realized by transferring the knowledge from the source domain (SD) to the target domain (TD). In general, DTL can be divided into three patterns: instance-based transfer, feature-based transfer, and parameter-based transfer (PTL) [21]. The first two methods assume that the samples or learned features of SD and TD have a similar distribution. Liu et al. [22] proposed selective multiple instance transfer learning, which measures the correlation between tasks in the source and target domains by investigating the similarity of features between two tasks. This method solves the problem of knowledge security transfer in multi-instance learning. Wang et al. [23] constructed a domain-adaptive transfer learning network by minimizing the maximum mean discrepancy between source and target domains to reduce marginal distribution bias. For the above methods, it is difficult to develop an algorithm with generalization ability to reduce the feature differences between different domains. In comparison, PTL has a wider range of applications and ensures user data privacy and security. In the PTL strategy, the feature extraction backbone of DL is applied to SD for training to obtain the pre-training backbone, and then the pre-training backbone containing the trained weight parameters is applied to TD. Data sharing is not involved in the process of knowledge transfer. Zhang et al. [24] proposed federated transfer learning based on prior distribution, which achieves local fault diagnosis for multiple users by uploading local models and downloading global models. Chen et al. [25] used a one-dimensional convolutional neural network (CNN) as the feature extraction backbone to implement parameter transfer to bearing and motor datasets. Wen et al. [26] used the image of the bearing vibration signal from the time domain as input to train the pre-training model of ResNet-50 from ImageNet [27]. Although CNN and its numerous variants have achieved great success in PTL, the important features cannot be considered due to the limitations of convolutional layers and uniform feature consideration. With the success of the vanilla transformer [28] in natural language processing (NLP) and computer vision (CV), it began to be used in DTL as an excellent backbone for feature extraction. Pei et al. [29] used a vanilla transformer as a feature extraction backbone and CNN as a classifier to improve fault diagnosis from multiple classes to a few classes on bearing and gearbox datasets, respectively. In the current study, the vanilla transformer is more successful in fault diagnosis with balanced data. Ding [30] combines time-frequency signal analysis with vanilla transformers to achieve fault diagnosis of bearing datasets by mining important features in time-frequency maps. Tang [31] uses a vision transformer (ViT) [32] to perform preliminary diagnosis on time-frequency maps of different frequency bands and fuse sub-results through the soft voting method to obtain the final diagnostic decision. Moreover, PTL fault diagnosis is more likely to be performed between different operating states of the same RM component, which is not possible for different RM components due to the large distance between domains.

The above methods are mainly used for a specific RM component, and there is no single method applicable to most RM components. Therefore, this work explores a paradigm that can apply fault diagnosis to multiple RM components based on imbalanced data. On the one hand, synchrosqueezed wavelet transforms (SWT) [33] are further improved in this work to compress the frequency scale of samples and obtain time-frequency characteristics of different samples under complex working conditions. On the other hand, a hierarchical window transformer pipeline obeying a dynamic seesaw (HWT-SS) has been designed to improve the feature extraction capability for imbalanced samples. The proposed methods are verified on two bearing datasets, one gearbox dataset, and one motor dataset to demonstrate their excellent performance. At the same time, the model realizes the visualization of attention by the weighted sum of key features by Grad-CAM [34], which improves the interpretability of the model. The main contributions of this work are as follows:(1)The improved SWT performs scale compression in the frequency dimension and normalizes the amplitude energy of the frequency. Thus, the difference between different components is reduced, and the most important features are represented more intensively in the time-frequency plane.(2)A novel transformer-based pipeline (HWT-SS) uses the hierarchical window transformer (HWT) as a backbone. The seesaw loss function is applied to realize the dynamic equilibrium of different classes of samples in the training process.(3)Cross-component transfer learning experiments (THWT-SS) on four datasets with multiple imbalanced ratio samples can effectively improve the accuracy and robustness of RM fault diagnosis with imbalanced data.

The remaining paper is organized as follows. Section 2 describes the background theory of the transformer backbone. The details of the proposed method are summarized in Section 3. Section 4 presents the experimental details of the proposed method using four datasets. Finally, the conclusions are presented in Section 5.

## 2. Primary Theories of Transformer Backbone

Currently, transformers and their variants are usually integrated into research tasks as the backbone of feature extraction. Moreover, the Swin transformer proposed by Liu et al. [35] has shown better performance than ViT on many visual tasks. The basic framework for feature extraction proposed in this work was inspired by the Swin transformer. In this section, we briefly review the basic theories of the Swin transformer in the context of this work.

### 2.1. Multi-Head Self-Attention

Self-attention (SA) is the heart of the transformer, which refers to the attention values of one vector over other vectors. Specifically, the input matrix vector X of the transformer is embedded to obtain queries Q, keys K , and values V vectors by three initialized weight matrices $W^{q}$, $W^{k}$ and $W^{v}$. The implementation formula is shown in Equation (1):(1)$$Q=Embedding(X)W^{q}\;K=Embedding(X)W^{k}\;V=Embedding(X)W^{v}$$

The attention weight between vectors is determined by scaling and softmax operations according to the dot product of the query vector and key vector. Then, a new vector containing the attention relationship with other time series vectors can be obtained by applying the attention weight to the value vector. The corresponding matrix calculation is shown in Equation (2), where $\sqrt{d_{k}}$ is the scaling factor and $d_{k}$ is the dimension of the keys vector.
(2)$$Z_{Attention}(Q_{i},K_{i},V_{i})=soft\max(\frac{Q_{i}K_{i}^{T}}{\sqrt{d_{k}}})V_{i}$$

Multi-head self-attention (MSA) was introduced to obtain better attention values between vectors, which means that the input vector X is embedded in n subspaces. Each subspace H performs an SA computation, and the results are combined by the trainable matrix $W^{O}$, The implementation formula is shown in Equation (3):(3)$$Z_{MultiHead}=Concat(Z_{Attention}^{H_{1}},Z_{Attention}^{H_{2}},\cdot \cdot \cdot ,Z_{Attention}^{H_{n}})W^{O}$$

### 2.2. Encoder Block

The encoder block is the basic module of HWT, which is composed of several function blocks. The structure of the encoder block includes a patch merging block and a stackable MSA block, as shown in Figure 1. Two types of MSA modules make up the attention mechanism block: Windows-based MSA (W-MSA) and shifted windows-based MSA (SW-MSA). Each MSA module is followed by a multi-layer perceptron (MLP) module, and the layer normalization (LN) and the residual connector are applied in each module.

In detail, the patch merging block with the down-sampling function reduces the feature size and increases the feature dimension of the embedded feature maps $Z^{E}$ by merging the corresponding positional features of each adjacent sliding block in the original feature maps, as shown in Equation (4):(4)$$Z^{l-1}=LN(Unfold(Z^{E}))$$
where the Unfold function is like $Conv_{2D}$ without convolution operation.

Then the feature maps $Z^{l-1}$ are input to the W-MSA module, which arranges the windows so that the feature maps are segmented evenly without overlapping and applies MSA to each window. The computational complexity is reduced by W-MSA compared to global MSA. The MLP module after W-MSA can improve the convergence of the model and prevent overfitting by Gaussian error linear units (GELU) activation [36] and the dropout layer. Two full connection layers (FC) are used to ensure that the input and output dimensions of the MLP are consistent, and the expression formula of the MLP is expressed as Equation (5):(5)$$Z_{MLP}=Dropout(FC_{2}(GELU(FC_{1}(Z_{LN}))))$$
the successive W-MSA and the MLP can be expressed as Equation (6):(6)$$\begin{array}{c} {\hat{Z}}^{l}=W-MSA(LN(Z^{l-1}))+Z^{l-1}\\ Z^{l}=MLP(LN({\hat{Z}}^{l}))+{\hat{Z}}^{l} \end{array}$$

Finally, the feature maps $Z^{l}$ are input to SW-MSA and re-spliced through the shifted windows so that the connections between the adjacent windows of W-MSA are obtained by MSA. SW-MSA and the following MLP are computed as in Equation (7). The entire attention mechanism module can be connected in series N to extract more features.
(7)$$\begin{array}{c} {\hat{Z}}^{l+1}=SW-MSA(LN(Z^{l}))+Z^{l}\\ Z^{l+1}=MLP(LN({\hat{Z}}^{l+1}))+{\hat{Z}}^{l+1} \end{array}$$

## 3. The Proposed Methods

The improved SWT and HWT-SS are creatively proposed in this work for the imbalanced data of various RM components, which are described in this section.

### 3.1. Improved Synchrosqueezed Wavelet Transforms

Time-frequency analysis is widely used to deal with non-stationary signals from RM under complex working conditions. The continuous wavelet transform (CWT) is one of the most typical methods of time-frequency analysis, in which wavelet windows with variable shapes are obtained by introducing a time scale factor a and a translation factor b based on the parent wavelet. Moreover, SWT [33] reorders and compresses the wavelet coefficients in the frequency direction based on CWT, which focuses more on the time-frequency plane. In this way, the synchronized transformation $T_{s}(w_{l},b)$ in a successive bin can be determined by $\left[w_{l}-\frac{1}{2}\Delta w,w_{l}+\frac{1}{2}\Delta w\right]$ at the center frequency $w_{l}$. The specific calculation formula is shown in Equation (8):(8)$$T_{s}(w_{l},b)=(\Delta w{)}^{-1}\sum_{a_{k}:|w(a_{k},b)-w_{l}|\leq \Delta w/2}W_{s}(a_{k},b)a_{k}^{-3/2}(\Delta a{)}_{k}$$
where $W_{s}(a_{k},b)$ represents the wavelet coefficients calculated by discrete time-scale $a_{k}$, $(\Delta a{)}_{k}=a_{k}-a_{k-1}$, $\Delta w=w_{l}-w_{l-1}$.

Data differences between multiple domains can be reduced by using the same data pre-processing method. Therefore, the logarithmic scaling factor and the standardization of the synchrosqueezed transformation based on the Morlet parent wavelet proposed in this paper allow further domain matching of multiple RM data. Specifically, the logarithmic scaling factor is applied to the center frequency $w_{l}$ of the SWT to achieve scaling compression in the frequency direction. Z-score normalization is applied to normalize the value of the synchronized transform $T_{s}$, which can be formulated in Equation (9):(9)$$\hat{T_{s}}(\hat{w_{l}},b)=(S_{T}\Delta w{)}^{-1}\left[\log_{2}\left(\sum_{a_{k}:|w(a_{k},b)-w_{l}|\leq \Delta w/2}W_{s}(a_{k},b)a_{k}^{-3/2}(\Delta a{)}_{k}\right)-\mu_{T}\right]$$
where $\mu_{T}$, $S_{T}$ are the mean and variance of all discrete wavelet coefficients.

From Figure 2, it can be seen that the improved SWT further compresses time-frequency features ${\hat{T}}_{s}({\hat{w}}_{l},b)$ compared to the standard SWT, which not only highlights the fault features of the signal but also reduces inter-domain differences for subsequent cross-domain diagnosis.

### 3.2. Hierarchical Window Transformer Pipeline Obeying Dynamic Seesaw

To further improve the feature-learning capability of the diagnostic model, a novel transformer-based pipeline consisting of several modules connected in series is proposed, as shown in Figure 3. The main functions of each module are as follows:(1)Embedding layer converts time-frequency images into feature maps that can be input into the model.(2)Transformer encoder layer (TEL) extracts features through hierarchical encoding modules.(3)Generalized mean pooling avoids excessive feature loss by automatically updating parameter $p_{k}$ during training.(4)Seesaw loss function can reduce the impact of data imbalances during model training.

More detailed details of each module are introduced in subsequent sections.

#### 3.2.1. Embedding Layer

Time-frequency images must be embedded in patches that can be applied to the transformer encoder layer via the embedded layer. First, the patch partition is applied to segment RGB images into non-overlapping patches. The characteristic feature of each patch is the splitting of the pixel’s RGB values from the original image. The shape of the time-frequency images used as input is $X^{E}\in R^{H\times W\times 3}$, where H, W denotes the size of the time-frequency images and 3 is the number of dimensions. Convolution with a kernel size of s×s and a step size of s is applied to segment the image into patches, and the shape of the patches becomes $Z^{E}\in R^{\frac{H}{s}\times \frac{W}{s}\times 3s^{2}}$. Then the patches can be converted to arbitrary dimensions C by the patch embedding layer, which is achieved by linear mapping. Finally, the output tensors are normalized by LN. The embedded layer can be formulated by Equation (10):(10)$$Z^{E}=LN(FC(Conv_{2D}(X^{E})))$$

#### 3.2.2. Transformer Encoder Layer

TEL consists of three types of encoder block stacks with different numbers of stackable MSA blocks that learn features from the embedding sequence patches. The structure of each encoder block is described in detail in Section 2. In the feedforward encoder block, the linear mapping between the encoder blocks is used to bisect the feature dimension, which is expressed by the formula shown in Equation (11). As the depth of the encoder block stack increases, the receptive field of the original feature map becomes larger through hierarchical feature extraction.
(11)$$Z_{i}^{l-1}=LN(FC(Z_{i-1}^{l+1}))i=1,2,3$$

#### 3.2.3. Generalized Mean Pooling

In general, the pooling layer is connected after the convolution operation of the CNN to aggregate the features and reduce the dimensions to avoid overfitting. The multidimensional feature maps that pass through the TEL also require a pooling operation. The two most typical pooling layers are max-pooling and average-pooling, where the collected features are lost to some extent. Therefore, the generalized mean pooling (GeM) [37] used in this work is a pooling operation with a learnable parameter $p_{k}p_{k}$ and the formula shown in Equation (12) is differentiable in backpropagation:(12)$$Z_{k}^{G}={\left(\frac{1}{\left|X_{k}^{G}\right|}\sum_{x\in X_{k}^{G}}x^{p_{k}}\right)}^{\frac{1}{p_{k}}}$$
which is generalized to average pooling and max pooling when $p_{k}=1$ and $p_{k}\to \infty$ respectively.

The feature maps $X^{G}\in R^{H\times W\times K}$ passing through the TEL are taken as input, whose $X_{k}^{G}$ denotes the k-th feature map of $X^{G}$ and x denotes the feature vectors of each feature map. The clamp function is applied as an activation to ensure x is greater than 0 and avoid the disappearance of the gradient. It replaces the rectified linear unit (ReLU) activation and is described in Equation (13): (13)$$x_{out}=\left\{\begin{array}{ccccc} \varepsilon_{min} & & x_{i} & \leq & \varepsilon_{min}\\ x_{i}, & \varepsilon_{min} & x_{i} & < & \varepsilon_{max}\\ \varepsilon_{max} & & x_{i} & > & \varepsilon_{max} \end{array}\right.$$
where $\varepsilon_{min}$ takes an infinite decimal number close to zero and $\varepsilon_{max}$ takes null. In this way, a $Z_{k}^{G}$ indicates $X_{k}^{G}$ processed by GeM, and the feature maps can be represented as vectors ${\left[Z_{1}^{G},\cdot \cdot \cdot ,Z_{K}^{G}\right]}^{T}$.

#### 3.2.4. Seesaw Loss Classifier

The classifier maps the learned features to the one-hot coding of the real labels. To achieve PTL between different domains, the classifier’s class number is set to the mutable parameter N. The feature vectors are linearly mapped to the predicted logits $z=\left[z_{1},\cdots ,z_{N}\right]$, and the probability $p_{i}$ that $z_{i}$ belongs to the class i encoded by one-hot is determined by softmax activation, given by Equation (14):(14)$$p_{i}=\frac{e^{z_{i}}}{\sum_{j=1}^{N}e^{z_{j}}}$$

In addition, an appropriate loss function is used to reduce the difference between the predicted and real labels. The cross-entropy (CE) loss function is commonly used. Considering the imbalanced data, this paper introduces the mitigation factor $M_{ij}$ and the compensation factor $C_{ij}$ into CE to build the seesaw loss function, the implementation formula is as shown in Equation (15):(15)$$\begin{array}{c} L\left(z\right)=-\sum_{i=1}^{N}y_{i}\log({\hat{p}}_{i})\\ {\hat{p}}_{i}=\frac{e^{z_{i}}}{\sum_{j\neq i}^{N}M_{ij}C_{ij}e^{z_{j}}+e^{z_{i}}} \end{array}$$
where $y_{i}$, ${\hat{p}}_{i}$ are the true probability and the modified prediction probability of the class ii, respectively. The mitigation factor $M_{ij}$ can be formulated by Equation (16), which refers to mitigating the negative gradient effects of the positive class i on the negative class j by a factor of ${\left(\frac{n_{j}}{n_{i}}\right)}^{\gamma }$, the instance numbers $n_{i}$, $n_{j}$ are accumulated at each training iteration and γ is a hyperparameter that can adjust the degree of mitigation.
(16)$$M_{ij}=\left\{\begin{array}{cc} 1, & n_{i}\leq n_{j}\\ {\left(\frac{n_{j}}{n_{i}}\right)}^{\gamma }, & n_{i}>n_{j} \end{array}\right.$$

However, the positive class i is incorrectly classified as a negative class j if the mitigation factor $M_{ij}$ is over-adjusted. Therefore, the compensation factor $C_{ij}$ is used to improve the penalty for misclassifying the sample, as shown in Equation (17), which works when the prediction probability $p_{j}$ of the negative class j is greater than that of the $p_{i}$ of positive class i,
(17)$$C_{ij}=\left\{\begin{array}{cc} 1, & p_{j}\leq p_{i}\\ {\left(\frac{p_{j}}{p_{i}}\right)}^{\lambda }, & p_{j}>p_{i} \end{array}\right.$$
where λ is a hyper-parameter that controls the adjustment scale. The process of implementing the Seesaw loss function is shown in Figure 4.

The detailed steps of the HWT-SS training process can be seen in Algorithm 1.
**Algorithm 1.** Training of HWT-SS.
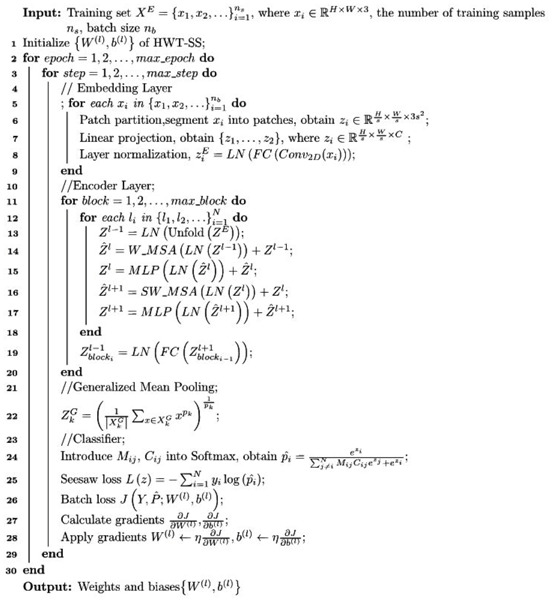


## 4. Experimental Study

The fault diagnosis performance of the model is verified and further analyzed in this section on four different datasets of RM components, including two types of bearing datasets, one gearbox dataset, and one motor dataset. There are two cases in which the performance of fault diagnosis is discussed:(1)Comparing with several classical DL models to check and analyze the performance of the HWT-SS in the presence of imbalanced data.(2)Performing THWT-SS between different datasets to verify and analyze the performance of PTL in cross-component fault diagnosis under imbalanced data. The framework is shown in Figure 5.

### 4.1. Datasets Description

Numerous validation tests were performed on four types of datasets, including the Case Western Reserve University (CWRU) bearing dataset [38], the Southeast University (SEU) gearbox dataset [39], the Shenzhen Technology University motor (SZTU-M) dataset, and the bearing (SZTU-B) dataset collected from the electromechanical fault test platform (PT650). The details of these datasets are described in this section.

#### 4.1.1. CWRU Bearing Dataset

The CWRU bearing dataset is one of the most commonly used datasets in performance verification of bearing fault diagnosis, which includes ball faults (BF), outer-race faults (ORF), and inner-race faults (IRF) caused by electro-discharge machining (EDM). These defects, ranging from 0.007 inches to 0.028 inches in diameter, were recorded from drive-end bearings SKF6205 with a sampling frequency of 12 kHz and 48 kHz, and fan-end bearings SKF6203 with a sampling frequency of 12 kHz, operating at constant motor speeds of 1720–1723 rpm for motor loads of 0–3 horsepower (hp). In addition, the healthy bearing test data for the normal condition (NC) is collected as a special bearing fault condition so that the CWRU dataset with ten bearing fault classes for four working conditions can be used.

#### 4.1.2. SEU Gearbox Dataset

SEU gearbox datasets are collected by the drivetrain dynamic simulator (DDS). The structure of DDS consists of a motor, a planetary gearbox, a parallel gearbox, and a brake. The gearbox dataset contains four types of faults and one state of condition, including chipping, miss, root, surface faults, and NC. Six channels of vibration signals are collected to describe each gear state under two types of rotational speed loads, including 20 Hz-0 V and 30 Hz-2 V. Therefore, the gear dataset of two working conditions with five gear fault classes can be used.

#### 4.1.3. SZTU Motor Dataset and Bearing Dataset

The motor dataset and the bearing dataset from Shenzhen Technology University (SZTU) are collected in our laboratory, as shown in Figure 6. The bearing and motor fault simulation experiment is conducted using the PT600, which consists of a three-phase asynchronous motor, two bearing pedestals, a planetary gearbox, a frequency converter, and a magnetic particle brake. The magnetic particle brake provides a torque load of 0–50 N∙m and the speed of the motor can be adjusted via the frequency converter in a range of 0–1750 rpm.

The motor dataset is collected by exchanging the motors with different fault types on the PT600. The specific motor types and their descriptions are summarized in Table 1. The working conditions are designed for variable speed and load, as shown in Table 2. Two triaxial accelerometers are installed on the drive end and fan end of the motor to acquire the vibration signal of the motor at a sampling rate of 51.2 kHz. The bearing dataset is collected by replacing bearing pedestals with various faults near the drive end of the motor. The models of the faulty bearings are UCPH206, and their more detailed description is shown in Table 3. The experiment is performed under two speed conditions with the four loads described in Table 4. Two triaxial accelerometers are installed on the top and side of the faulty bearing pedestal to collect vibration signals with a sampling rate of 40.96 kHz.

### 4.2. Experimental Setup

All working conditions in each dataset are mixed to verify the diagnostic effect of the proposed methods for complex working conditions. From each class of each dataset, 500 samples were randomly selected and divided into a training set, a validation set, and a test set in a ratio of 0.6:0.2:0.2. Since the sampling rate and data acquisition time are different, the sampling format of the datasets must be unified, considering the complete fault information. The length of each sample is set according to the sampling rate and speed of each dataset so that sample points of more than two rotation periods can be obtained. The specific descriptions of the four datasets used can be found in Table 5.

According to the PTL, the SD datasets $D_{i}^{s}$ (i=1,2,3,4) are original datasets with balanced samples, and the TD datasets $D_{j}^{t_{k}}$ (j=1,2,3,4,k=1,2,3) include imbalanced training samples created by randomly selecting fault samples in three imbalanced ratios (2:1, 10:1, 50:1), and balanced validation and test samples that are the same as the source domain. The imbalanced ratio refers to the ratio of NC samples to fault samples in each dataset. Further details are described in Table 6.

Two experimental cases are conducted. Case 1: In four datasets with different data ratios, ResNet and VGG are used as benchmark models to compare with the proposed HWT-SS. Meanwhile, HWT using the CE loss function (HWT-CE) is used to verify the performance of the Seesaw loss function. Case 2: Application of PTL between different datasets. $D_{i}^{s}\to D_{j}^{t_{k}}$ represents that the weight parameters are learned by pre-training in the $D_{i}^{s}$ and then applied to the $D_{j}^{t_{k}}$ while further training.

To ensure the fairness of all comparison experiments, the training parameters of the model must be standardized before training. In addition, Adam with decoupled weight decay (AdamW) [40] is used as an optimizer to prevent overfitting. All one-dimensional vibration signal samples are converted into images by improved SWT processing before training. The training hyperparameters and network structure parameters of the proposed HWT-SS are described in Table 7. The details of the benchmark models are described in Table 8. Meanwhile, all experiments were conducted in the same computing environment, including AMD Ryzen 7-5800, NVIDIA GeForce RTX 3070Ti with 8 GB of memory, CUDA 11.3, and the Pytorch 1.10.1 framework.

### 4.3. Results Analysis and Comparisons

#### 4.3.1. Case 1: Fault Diagnosis of Four Datasets with Different Imbalanced Ratios

We compared HWT-SS with HWT-CE, VGG, and ResNet on four datasets with different imbalances and repeated the training verification five times to show the robust generalization ability of the model. Figure 7 illustrates the comparison of diagnostic accuracy and standard deviation for the four datasets. From a macroscopic point of view, the average accuracy and standard deviation of each model at different imbalanced ratios of the four datasets are shown in Table 9. On the one hand, the accuracy of HWT-SS at the balanced ratio (1:1) is basically the same as that of HWT-CE, which is 99.95% ± 0.03% and 99.88% ± 0.09%, respectively. The advantage of the seesaw loss function becomes more obvious with the increase in the imbalanced ratio, which increases the accuracy by 0.33%, 2.3%, and 3.75% compared with HWT-CE. At the same time, the stability of HWT-SS was also improved in terms of standard deviation. On the other hand, the average accuracy of HWT-SS on different data ratios was improved by 1.32%, 2.47%, 7.86%, and 11.29% compared to ResNet, and the performance benefits of HWT-SS were increased by 2.48%, 4.24%, 11.48% and 14.9% compared to VGG, respectively, due to hierarchical feature extraction and dynamic compensation of the seesaw loss function for imbalanced data.

To further explore the diagnostic capability of the proposed HWT-SS for imbalanced data, the low imbalanced ratio of 2:1 and the high imbalanced ratio of 50:1 are highlighted. The t-distributed stochastic neighborhood embedding (T-SNE) [41] is applied to the last hidden layer of the model, and the high-dimensional features can be simplified to a two-dimensional distribution labeled in terms of prediction classes. Taking the CWRU dataset with the most fault classes as an example, Figure 8 shows the T-SNE visualization results of each model when the imbalance is 2:1. The analysis results in Figure 8a illustrate that the feature vectors obtained from HWT-SS have the best intra-class aggregation and inter-class separability, while the intra-class aggregation of the other three models is poor. At the same time, we note that the T-SNE visualization of each model leads to different degrees of misclassification, which is due to the influence of the NC samples. To further quantify the misclassification of the different models, the confusion matrixes are used to represent the prediction accuracy of each class, as shown in Figure 9. At the same time, the NC precision is calculated to investigate the extent to which fault samples are misclassified as NC samples, and the formula is presented in Equation (18):(18)$$P=\frac{TP}{TP+FP}\times \%$$
where TP and FP represent the number of true and false NC samples.

As can be seen in Figure 9a, HWT-SS has the lowest misclassification; only three ORF7 samples are misclassified as NC, and the accuracy of HWT-SS for the NC label is at most 97.09%. In HWT-CE and ResNet, there are 7 and 38 ORF7 samples misclassified as NC labels, respectively (see Figure 9b,c), and the NC accuracy is 92.59% and 72.46%, respectively. VGG performed the worst with 51 and 49 prediction errors for NC and ORF7, respectively.

In addition, Figure 10 shows the T-SNE visualization of each model in an extremely imbalanced dataset $D_{1}^{t_{3}}$ where there are only 6 samples for each fault class. The proposed HWT-SS still has the best inter-class separability and intra-class aggregation, and the class features of the other three models are blended with each other. The class predictions of the different models for the dataset $D_{1}^{t_{3}}$ are shown in Figure 11 using the confusion matrix. The prediction effect of HWT-SS for each class is still better than that of the other three models; only a small number of prediction faults exist for the IRF14 and OPF7 labels. But with the increase in the ratio of normal samples to fault samples, the NC precision of HWT-SS decreases to 66.23%.

Consequently, HWT-SS has the best performance compared to the other three models, and the performance advantage is more obvious the larger the imbalanced ratio. However, the accuracy and stability of HWT-SS have decreased with extremely imbalanced data, so PTL is used to further improve the diagnostic performance.

#### 4.3.2. Case 2: Cross-Component Fault Diagnosis with PTL

Since the accuracy of HWT-SS is 97.28% even with the extreme imbalance of the SZTU-M dataset, we set three extremely imbalanced datasets $D_{1}^{t_{3}}$, $D_{2}^{t_{3}}$, $D_{4}^{t_{3}}$ other than the SZTU-M dataset as TD, and four normal datasets $D_{1}^{s}$, $D_{2}^{s}$, $D_{3}^{s}$, $D_{4}^{s}$ as SD for cross-component fault diagnosis. Thus, nine groups of PTL tests were performed. The accuracy of each test group and the average accuracy of each TD are shown in Table 10. The following conclusions can be drawn from the comparative analysis:(1)It is not difficult to see that different SD datasets have different transfer effects on the TD. Specifically, $D_{3}^{s}\to D_{1}^{t_{3}}$ test has the best result of transfer learning, whose diagnostic accuracy is 99.06% ± 0.63%. For the number of fault classes, $D_{3}^{s}$ is more similar to $D_{1}^{t_{3}}$ than other SD datasets.(2)When the number of fault classes in the different SDs is similar, the transfer learning effect is better when the sample length of SD is similar to that of TD, which is confirmed by the accuracy of $D_{2}^{s}\to D_{1}^{t_{3}}$ (95.04% ± 0.61%), which is greater than the accuracy of $D_{4}^{s}\to D_{1}^{t_{3}}$ (93.14% ± 0.79%).

Similarly, the above analysis can be confirmed for the $D_{2}^{t_{3}}$ and $D_{4}^{t_{3}}$ tests. Since $D_{1}^{s}$ and $D_{3}^{s}$ have more fault types, the accuracy of $D_{1}^{s}\to D_{2}^{t_{3}}$ and $D_{3}^{s}\to D_{2}^{t_{3}}$ is significantly higher than that of $D_{4}^{s}\to D_{2}^{t_{3}}$, and the accuracy of $D_{1}^{s}\to D_{4}^{t_{3}}$ and $D_{3}^{s}\to D_{4}^{t_{3}}$ are significantly higher than that of $D_{2}^{s}\to D_{4}^{t_{3}}$. Due to the SD and the TD of $D_{1}^{s}\to D_{2}^{t_{3}}$ and $D_{3}^{s}\to D_{4}^{t_{3}}$ transfer tests have similar sample lengths, their accuracy is also the highest in their group of TD tests. Surprisingly, due to the limitations in the number of fault classes and sample length in the SD, transfer learning between components of the same type did not achieve the best result, which can be seen from the accuracy of the bearing component transfer tests in the $D_{4}^{s}\to D_{2}^{t_{3}}$ and $D_{2}^{s}\to D_{4}^{t_{3}}$. When the number of fault classes included in the source domain is large, TD can learn more feature information from SD. If the sample length of SD and TD is similar, which means the rotation speed and sampling rate are similar. Then the state of the components in SD and TD are similar, so the state information can be transferred and learned better.

To further explore the advantages of THWT-SS compared to the models in Case 1, we also take the CWRU dataset as an example. First, the accuracy and loss curves of the training process for each model are shown in Figure 12. THWT-SS is stable after 43 epochs, and its performance can quickly converge to the high accuracy region compared to the models before 10 epochs. From the T-SNE visualization of three transfer tests in Figure 13, THWT-SS performs better in intra-class aggregation and inter-class separation of different feature classes than Figure 10, where $D_{3}^{s}\to D_{1}^{t_{3}}$ obviously performs the best. The class prediction of each transfer test is shown in Figure 14 using the confusion matrix. The NC precision increased from 66.23% to 92.59% for the maximum extent, and the accuracy of the other classes also improved further compared with Figure 11a.

In addition, Grad-CAM is used to improve the interpretability of the THWT by drawing attention to the characteristics of the sample. For a test image, the gradient information for the target class is propagated back to a feature layer of the model, which becomes Grad-CAM visualization through weighted summation and ReLU activation. From the confusion matrixes in Figure 11 and Figure 14, the classes affected by the extremely imbalanced data in the CWRU dataset are mainly IRF14 and ORF7. Therefore, the Grad-CAM heatmaps of the NC, IRF14, and ORF7 samples for the last feature layer of the HWT-SS and the THWT-SS ($D_{3}^{s}\to D_{1}^{t_{3}}$) are shown in Figure 15. The red area is the attention of the sample features compared to the original samples. It is not difficult to see that the focus of HWT-SS has evolved with the PTL towards a more precise and smaller scale, resulting in the model being able to better identify different classes.

We continue to test all imbalanced ratios in the CWRU, SEU, and SZTU-B datasets. The average accuracy of the transfer tests for each imbalanced ratio is calculated and compared to other models. From the comparison results shown in Figure 16, it is clear that THWT-SS has advantages over other models without PTL, which proves the feasibility of mutual transfer learning between different RM components. As a special case, the proposed HWT-SS is sufficient to handle various imbalanced ratios in the SZTU-M dataset, reflecting the advantage of the seesaw loss function in handling imbalanced data.

Finally, it is necessary to further evaluate the diagnostic performance of the proposed method compared with the published methods; the comparison results running under the same CWRU dataset are listed in Table 11. All models were compared under similar imbalanced ratios: Refs. [42,43] contain 12 fault classes under a single operating condition; Ref. [44] only has 5 fault classes under a single operating condition; and Ref. [45] has 10 fault classes under three mixed operating conditions. From the perspective of model diagnostic accuracy, the proposed THWT-SS achieves the highest accuracy of 100% under the most complex operating conditions. HWT-SS is only 0.04% worse than Ref. [45], ranking third, but it has an additional operating condition, which reflects the powerful performance of the seesaw loss function used in this paper.

## 5. Conclusions

In this paper, we proposed a novel THWT-SS to achieve fault diagnosis of RM with imbalanced data, which is composed of applying PTL to HWT-SS. The proposed THWT-SS has the following features: (1) We creatively apply the PTL to various RM components to solve the practical problem of RM and adopt the improved SWT to improve the signal feature expression in the time-frequency domain and reduce the feature difference between different domains. (2) The proposed HWT-SS adopts a hierarchical window transformer as the feature extraction backbone and dynamic seesaw loss as the loss function, which improves the feature extraction ability and reduces the impact of imbalanced data.

The advantages of the model are verified using two public and two self-generated datasets, respectively. First, in Case 1, the average accuracy of HWT-SS was increased by 3.75%, 11.29%, and 14.9% under the extreme imbalance condition compared to HWT-CE, ResNet, and VGG, respectively. In Case 2, the highest diagnostic accuracy of THWT-SS can reach more than 99.06% by transferring learning between different component datasets under an extremely imbalanced ratio condition. The comparisons with benchmark models and published methods prove that THWT-SS can solve the problem of RM imbalanced data by cross-component transfer learning. However, the model proposed in this paper still requires a small number of fault samples to complete fault diagnosis. In future work, we will further improve the model to solve the more extreme imbalanced problem and further explore the application of the model in domains other than RM, such as electrical appliances.

## Figures and Tables

**Figure 1 sensors-23-07431-f001:**
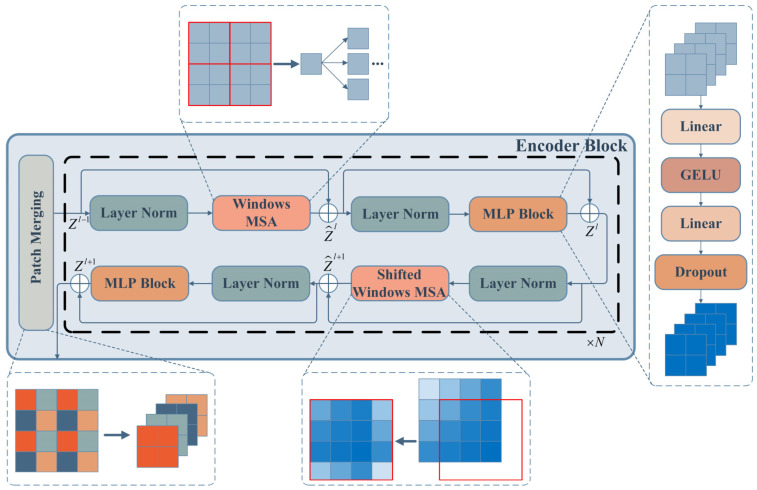
Swin transformer encoder block. Schematic diagram of features with different dimensions in different color boxes in patch merging block, the red lined boxes in W-MSA module is the windows for feature segmentation, the red lined boxes in SW-MSA module is a sliding window for feature recombination.

**Figure 2 sensors-23-07431-f002:**
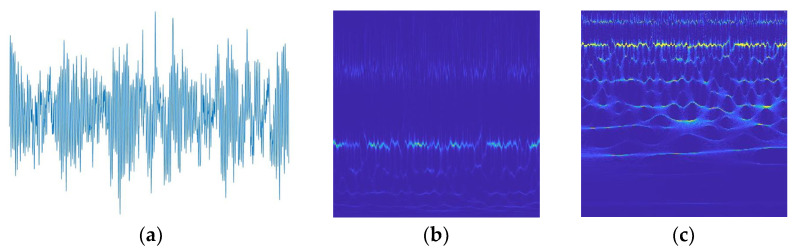
Improved SWT implementation process. (**a**) is the time domain plane of the vibration signal, (**b**) is the time-frequency plane according to the standard SWT, and (**c**) is the time-frequency plane according to the improved SWT proposed in this work.

**Figure 3 sensors-23-07431-f003:**
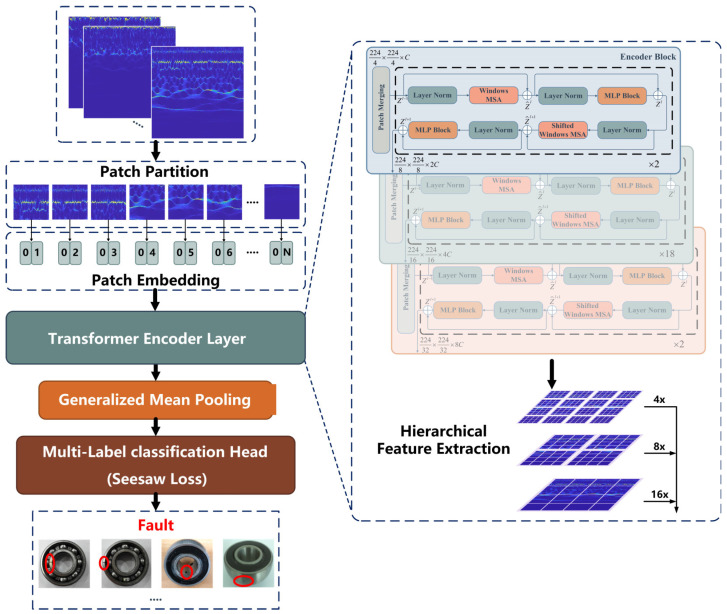
Structure of the HWT-SS pipeline. The red circles represents the fault location.

**Figure 4 sensors-23-07431-f004:**
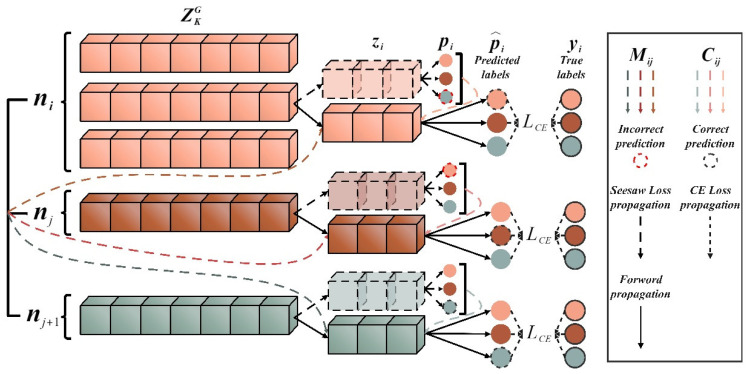
The Seesaw loss function implementation process.

**Figure 5 sensors-23-07431-f005:**
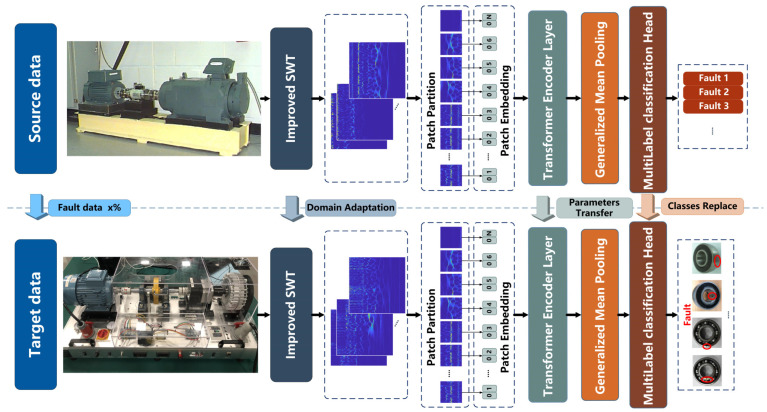
The PTL framework proposed in this paper. The red circles represents the fault location.

**Figure 6 sensors-23-07431-f006:**
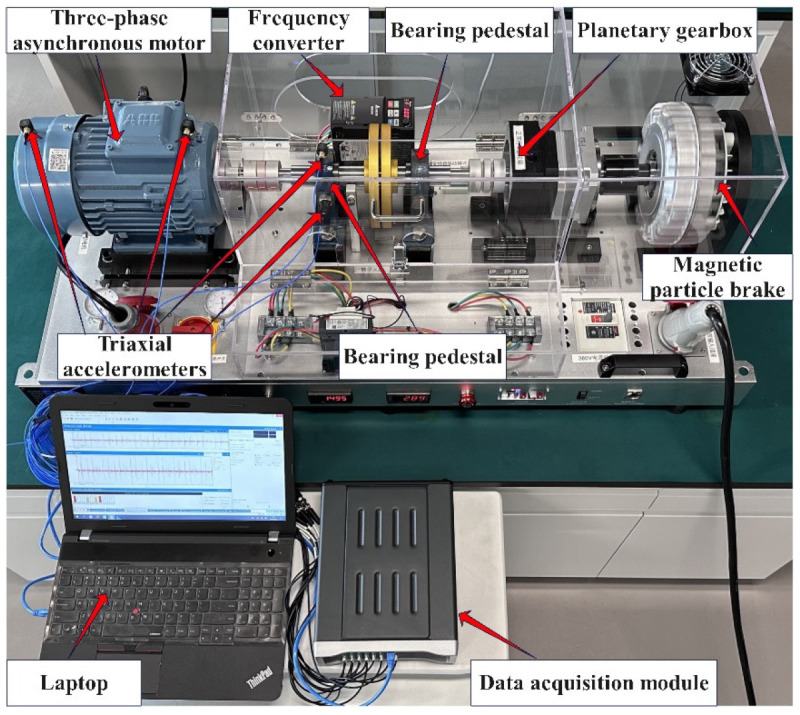
Electromechanical fault test rig PT600.

**Figure 7 sensors-23-07431-f007:**
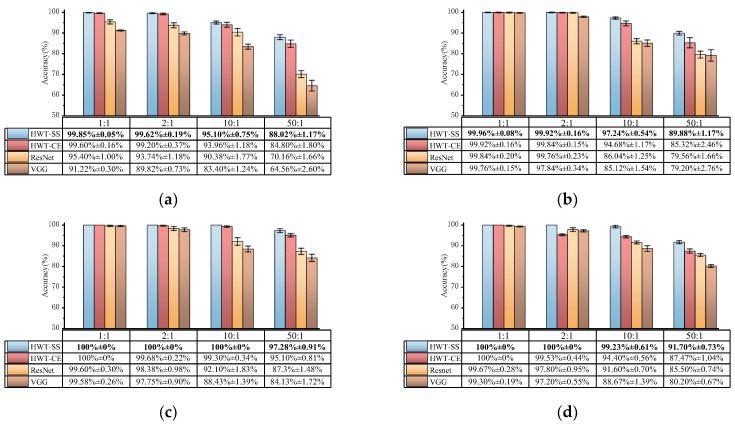
Fault diagnosis results of different models on the (**a**) CWRU, (**b**) SEU, (**c**) SZTU-M, (**d**) SZTU-B datasets with different imbalanced data ratios.

**Figure 8 sensors-23-07431-f008:**
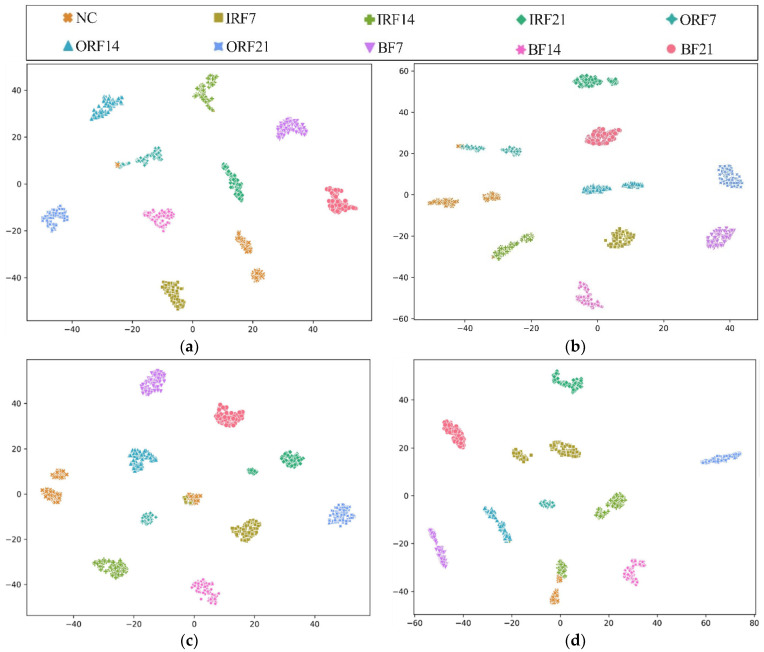
Visualizations of feature vectors extracted from the CWRU test dataset $D_{1}^{t_{1}}$ using (**a**) HWT-SS, (**b**) HWT-CE, (**c**) ResNet, and (**d**) VGG.

**Figure 9 sensors-23-07431-f009:**
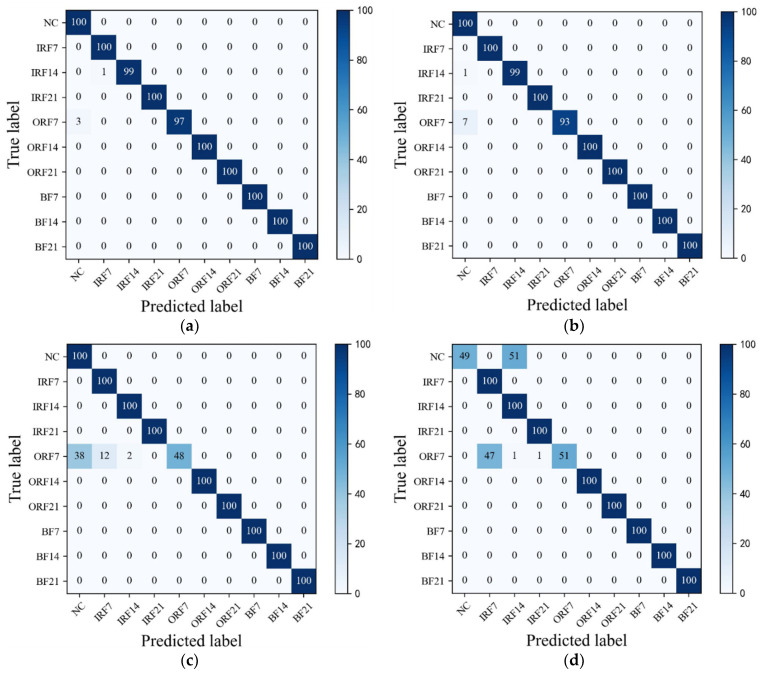
The confusion matrixes of (**a**) HWT-SS, (**b**) HWT-CE, (**c**) ResNet, and (**d**) VGG under the CWRU test dataset $D_{1}^{t_{1}}$.

**Figure 10 sensors-23-07431-f010:**
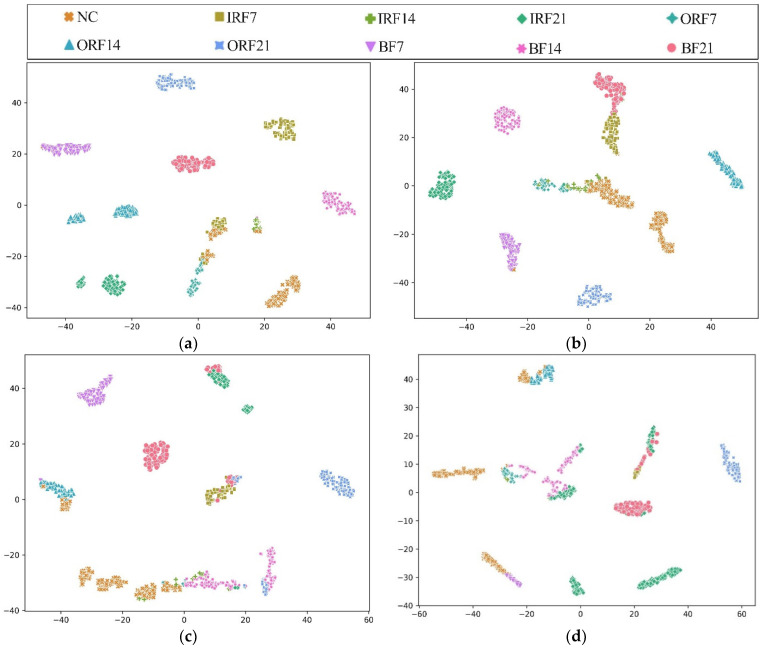
Visualizations of the feature vectors extracted from the CWRU test dataset $D_{1}^{t_{3}}$ using (**a**) HWT-SS, (**b**) HWT-CE, (**c**) ResNet, and (**d**) VGG.

**Figure 11 sensors-23-07431-f011:**
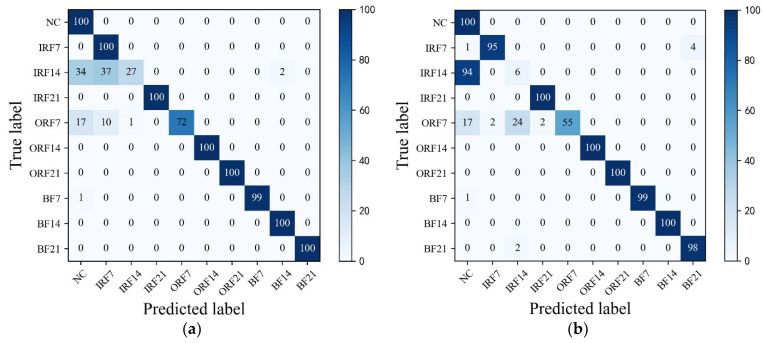
The confusion matrixes of (**a**) HWT-SS, (**b**) HWT-CE, (**c**) ResNet, and (**d**) VGG under the CWRU test dataset $D_{1}^{t_{3}}$.

**Figure 12 sensors-23-07431-f012:**
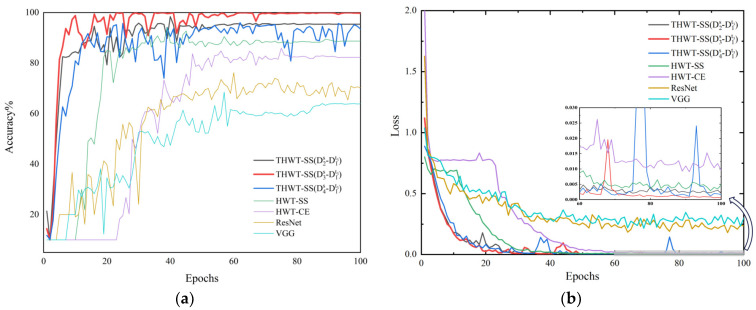
(**a**) is the training accuracy curve of each model, (**b**) is the training loss curve of each model.

**Figure 13 sensors-23-07431-f013:**
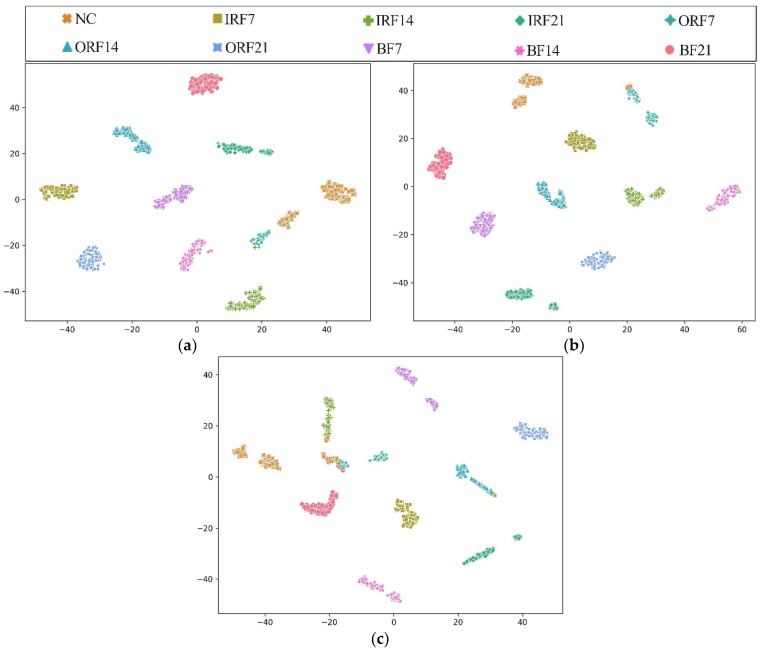
T-SNE visualization results of (**a**) $D_{2}^{s}\to D_{1}^{t_{3}}$, (**b**) $D_{3}^{s}\to D_{1}^{t_{3}}$, (**c**) $D_{4}^{s}\to D_{1}^{t_{3}}$ transfer tests across the THWT-SS.

**Figure 14 sensors-23-07431-f014:**
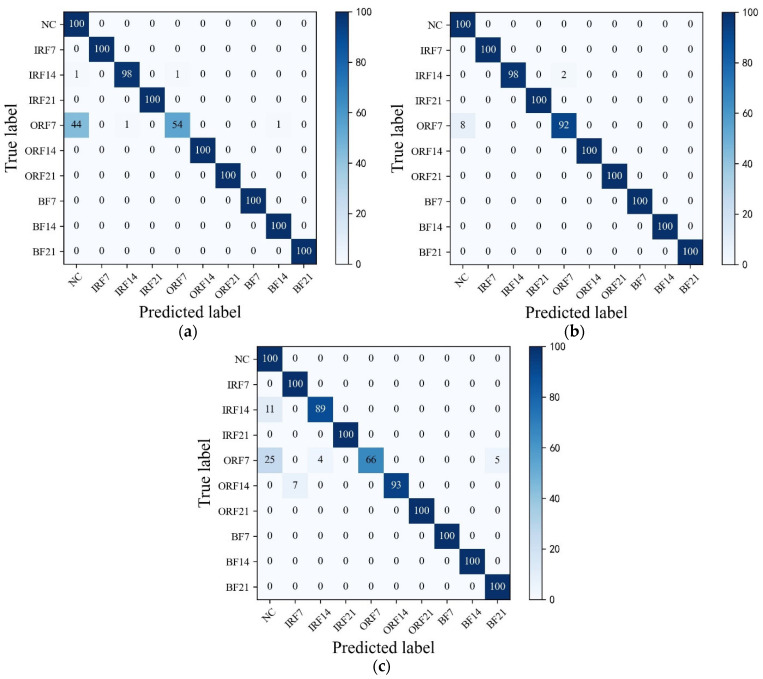
The confusion matrixes of (**a**) $D_{2}^{s}\to D_{1}^{t_{3}}$, (**b**) $D_{3}^{s}\to D_{1}^{t_{3}}$, (**c**) $D_{4}^{s}\to D_{1}^{t_{3}}$ transfer tests over the THWT-SS.

**Figure 15 sensors-23-07431-f015:**
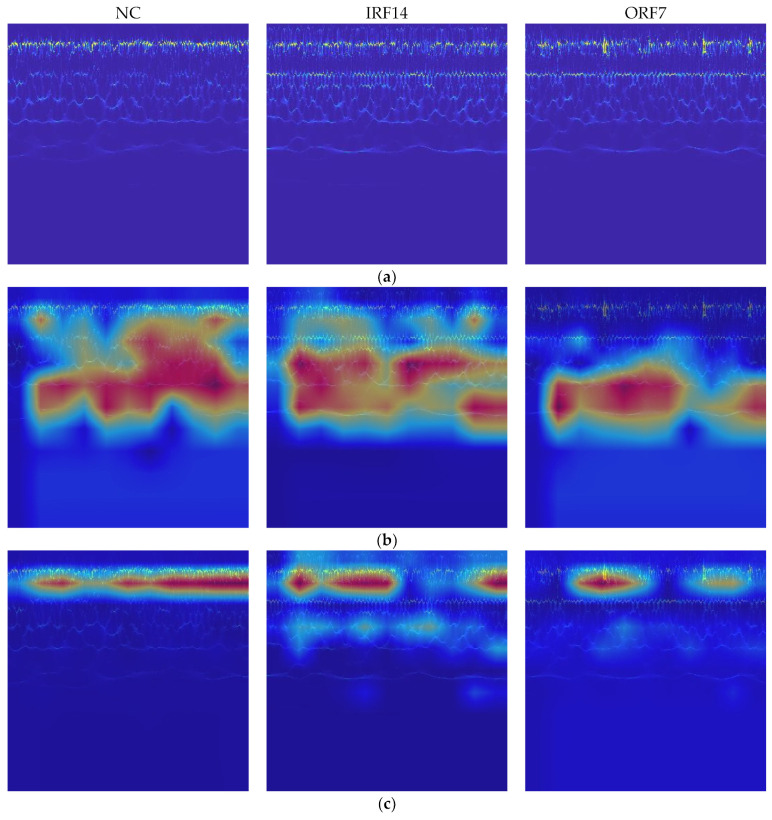
Grad-CAM heatmaps of (**a**) original samples, (**b**) HWT-SS and (**c**) THWT-SS ($D_{3}^{s}\to D_{1}^{t_{3}}$) model for NC, IRF14 and ORF7 label samples.

**Figure 16 sensors-23-07431-f016:**
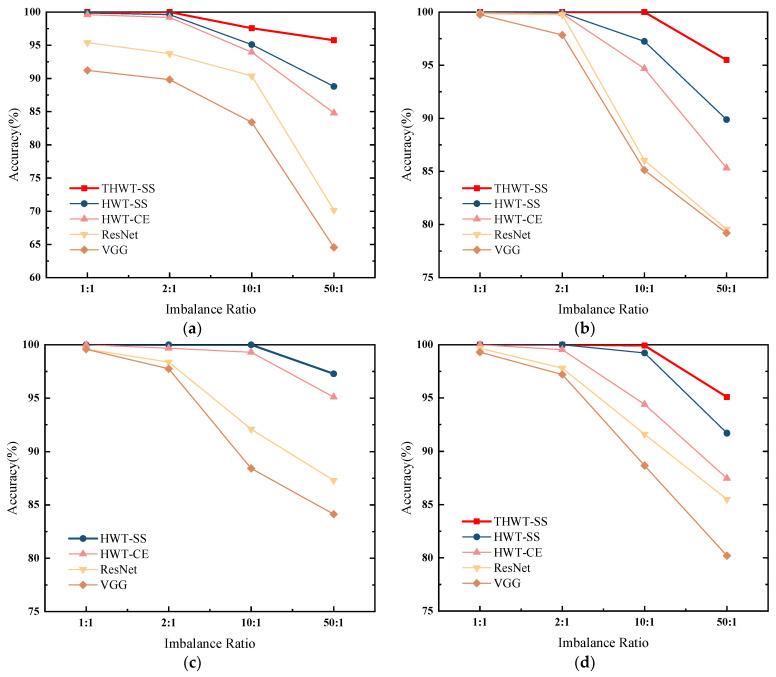
Comparative results of the individual models with different imbalance ratios on (**a**) CWRU, (**b**) SEU, (**c**) SZTU-M, (**d**) SZTU-B.

**Table 1 sensors-23-07431-t001:** Motor type description.

	Motor Type	Description
NRM	Normal motor	Healthy motor without defects
RUM	Rotor imbalanced motor	The imbalanced is caused by adding 14 g of imbalanced mass to both ends of the rotor
RMM	Rotor misalignment motor	Precession in the jack bolt of the motor end cover by 0.5 mm
BRM	Bending rotor motor	Bending amount at both ends of the rotor shaft is 0.4 mm
FBM	Faulty bearing motor	The inner ring of the inner bearing and the outer ring of the outer bearing have cracks, the width is 0.35 mm and the depth is 3 mm
BBM	Broken bar motor	4 rotor bars are cut from 28 rotor bars
WFM	Stator winding fault motor	The stator winding has a short-circuit condition
SPM	Single phase fault motor	One of the three-phase windings is disconnected

**Table 2 sensors-23-07431-t002:** Motor dataset conditions.

Condition	Speed (rpm)	Load (N∙m)
1	1722	33
2	1490	17
3	1740	17
4	875	33

**Table 3 sensors-23-07431-t003:** Bearing type description.

	Bearing Type	Description
NRB	Normal bearing	Healthy bearing without defects
FBI	Inner ring fault bearing	The inner ring has a crack 0.35 mm wide and 2 mm deep
FBO	Out ring fault bearing	The out ring has a crack 0.35 mm wide and 2 mm deep
FBB	Ball fault bearing	The ball is drilled with a hole with a diameter of 0.5 mm and a depth of 2 mm
FBC	Compound fault bearing	There are 0.35 mm wide and 2 mm deep cracks on the inner and outer rings
FBR	Retainer fault bearing	Broken bearing retainer

**Table 4 sensors-23-07431-t004:** Bearing dataset conditions.

Condition	Speed (rpm)	Load (N∙m)
1	1200	0
2	1190	17
3	1174	33
4	1158	50

**Table 5 sensors-23-07431-t005:** Description of the four datasets.

	Dataset	Class	Class Number	Condition	Sample Length
$D_{1}^{s}$	CWRU	NC, IRF7, IRF14, IRF21, ORF7, ORF14, ORF21, BF7, BF14, BF21	10	12 kHz (Drive-end)-0&1&2&3 hp	1024
$D_{2}^{s}$	SEU	NC, Chipped, Miss, Root, Surface	5	20 Hz-0 V&30 Hz-2 V	1024
$D_{3}^{s}$	SZTU-M	NRM, RUM, RMM, BRM, FBM, BBM, WFM, SPM	8	1&3	3420
$D_{4}^{s}$	SZTU-B	NRB, FBI, FBO, FBB, FBC, FBR	6	1&2&3&4	4800

**Table 6 sensors-23-07431-t006:** Description of sample quantity in the SD and the TD of four datasets.

Dataset	Source Domain	Training Target Domain Normal/Fault			Validation/Test Target Domain Normal/Fault		
	$D_{j}^{s}$	$D_{j}^{t_{1}}$(2:1)	$D_{j}^{t_{2}}$(10:1)	$D_{j}^{t_{3}}$(50:1)	$D_{j}^{t_{1}}$(2:1)	$D_{j}^{t_{2}}$(10:1)	$D_{j}^{t_{3}}$(50:1)
CWRU	500 × 10	300 × 1/150 × 9	300 × 1/30 × 9	300 × 1/6 × 9	100 × 10/100 × 10		
SEU	500 × 5	300 × 1/150 × 4	300 × 1/30 × 4	300 × 1/6 × 4	100 × 5/100 × 5		
SZTU-M	500 × 8	300 × 1/150 × 7	300 × 1/30 × 7	300 × 1/6 × 7	100 × 8/100 × 8		
SZTU-B	500 × 6	300 × 1/150 × 5	300 × 1/30 × 5	300 × 1/6 × 5	100 × 6/100 × 6		

**Table 7 sensors-23-07431-t007:** Hyperparameters and structure of the proposed HWT-SS.

Hyperparameters and Structure	Value
Input size	224 × 224 × 3
Batch size	10
Max epochs	100
GeM pooling $p_{k}$/$\varepsilon_{min}$	3/1 × 10^−6^
Seesaw loss γ/λ	0.8/2
Drop rate	0.3
AdamW learning rate/weight-decay	5 × 10^−5^/0.05
Embedding layer s	4
Number of MSA in encoder block 1	2
Output feature dimension of encoder block 1	192
Number of MSA in encoder block 2	18
Output feature dimension of encoder block 2	384
Number of MSA in encoder block 3	2
Output feature dimension of encoder block 3	768

**Table 8 sensors-23-07431-t008:** Hyperparameters and structure of the benchmark models.

Model	Hyperparameters and Structure	Value
ResNet	Input size	224 × 224 × 3
	Batch size	16
	Max epochs	100
	Structure	ResNet18
	Drop rate	0.3
	Adam learning rate	5 × 10^−5^
	Pooling layer	Global average pooling
	Loss function	Cross-entropy loss
VGG	Input size	224 × 224 × 3
	Batch size	16
	Max epochs	100
	Structure	VGG11
	Drop rate	0.3
	Adam learning rate	5 × 10^−5^
	Pooling layer	Global average pooling
	Loss function	Cross-entropy loss

**Table 9 sensors-23-07431-t009:** The average accuracy and standard deviation of the models in four datasets.

	1:1	2:1	10:1	50:1
HWT-SS	99.95% ± 0.03%	99.89% ± 0.09%	97.89% ± 0.48%	91.92% ± 0.74%
HWT-CE	99.88% ± 0.08%	99.56% ± 0.30%	95.59% ± 0.81%	88.17% ± 1.53%
ResNet	98.63% ± 0.23%	97.42% ± 0.63%	90.03% ± 1.39%	80.63% ± 1.94%
VGG	97.47% ± 0.45%	95.65% ± 0.84%	86.41% ± 1.39%	77.02% ± 1.39%

**Table 10 sensors-23-07431-t010:** Cross-component fault diagnosis results with PTL.

	$D_{1}^{s}$	$D_{2}^{s}$	$D_{3}^{s}$	$D_{4}^{s}$	Average Accuracy
$D_{1}^{t_{3}}$	/	95.04% ± 0.61%	99.06% ± 0.63%	93.14% ± 0.79%	95.75% ± 0.68%
$D_{2}^{t_{3}}$	97.32% ± 0.37%	/	96.80% ± 0.49%	91.04% ± 0.98%	95.49% ± 0.61%
$D_{4}^{t_{3}}$	96.47% ± 0.54%	90.23% ± 0.36%	97.92% ± 0.61%	/	95.08% ± 0.50%

**Table 11 sensors-23-07431-t011:** Comparison results of different models.

Method	Class Number	Condition	Imbalanced Ratios (NC:BF:IF:OF)	Accuracy	Rank
THWT-SS	10	0&1&2&3 hp	2:1:1:1	100%	1
HWT-SS	10	0&1&2&3 hp	2:1:1:1	99.62%	3
HWT-CE	10	0&1&2&3 hp	2:1:1:1	99.20%	4
ResNet	10	0&1&2&3 hp	2:1:1:1	93.74%	9
VGG	10	0&1&2&3 hp	2:1:1:1	89.82%	8
Ref. [42]	12	1 hp	7:1:3:5	95.36%	6
Ref. [45]	10	1&2&3 hp	10:5:8:2	99.66%	2
Ref. [44]	5	0 hp	2:1:1	94.85%	7
Ref. [43]	12	3 hp	2:1:1:1	96.80%	5

## Data Availability

Data available on request due to restrictions.

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
