# Peer review of "Cross-Component Transferable Transformer Pipeline Obeying Dynamic Seesaw for Rotating Machinery with Imbalanced Data"

_sensors, 2023, doi:10.3390/s23177431_

Round 1

Reviewer 1 Report

The authors propose a novel hierarchical window transformer model that obeys a dynamic seesaw (HWT-SS), and a creative strategy of transfer diagnosis between multiple rotating mechanical components combined with transfer learning is proposed. It is a well-structured paper with interesting results. However, it requires further improvements.

(1)The abstract should be improved. Your point is your own work that should be further highlighted.

(2)The parameters in expressions are given and explained.

(3) The method in the context of the proposed work should be written in detail.

(4) In the Section 3 of The proposed methods, the idea of the proposed method is not clear.

(5) In Section 3.2, "Error! Reference source not found.." What's mean?

(6) The article can be further enhanced by connecting the undergoing work with some existing literatures. https://doi.org/10.3390/rs15133402; http://dx.doi.org/10.1145/3513263; http://dx.doi.org/10.1145/3513263 and so on.

(7) What are the limitations behind this study? This topic should be highlighted in the Conclusion of manuscript.

Extensive editing of English language required

Reviewer 2 Report

This paper proposes Cross-component transferable transformer pipeline obeying dynamic seesaw for rotating machinery with imbalanced data. In general, this paper is well presented. The following issues can be further considered.

1. More background and motivation of this study can be added, in case the readers are not very familiar with the topic.

2. The descriptions of the well known knowledge can be properly reduced.

3. Why introducing the transformer based method for the problem? What is the major benefits compared with traditional methods?

4. Some related works on this topic should be reviewed, such as "Data privacy preserving federated transfer learning in machinery fault diagnostics using prior distributions","Intelligent Machinery Fault Diagnosis With Event-Based Camera", etc.

5. A couple of ablation studies should be added to evaluate the effects of the key parameters of the proposed method on the performance.

Reviewer 3 Report

The paper discusses the modification of the Swin Transformer to diagnose faults in rotating machinery in the case of imbalanced data. It corresponds to the aims and scope of the journal "Sensors".

There are a few major comments:

1. Abbreviations (see, for example, "CNC machines") must be explained immediately when they first appear in the text.

2. Empty issues (see pp. 27-28) should be excluded.

3. In the first sentence of subsection 2.1, the fonts for the values K, V, etc. are strange. The same problem continues through the text for other quantities.

4. Subsection 2.2 is numbered as 2.1.

5. There are multiple errors in the text: "Error! Reference source not found". The text of the manuscript, especially in the section with results, is fully unreadable.

6. The content of section 4.1.3 should be placed in the appendix to the manuscript.

7. How were the hyperparameter values selected in Tables 7 and 8? Are these optimal values and by what criterion?

There are no additional comments.

Round 2

Reviewer 1 Report

This paper can be accepted now.

Reviewer 2 Report

Comments are addressed. It can be accepted

Reviewer 3 Report

Now I can recommend acceptance in the current form.